Genome-wide identification and expression analysis of the HSP70 gene family in Artemisia annua L. under heat stress

Zhong Shan 1
Pan Hengyu 2
Ma Chaoxue 1
Xu Haojia 1
Ding Xiaoxia 2 3
Bao Shengye 2
Zhao Siyu 2
http://orcid.org/0000-0002-9535-5300 Shi Peiqi 2
Liao Baosheng 2 liaobaosheng@gzucm.edu.cn
Zong Xianchun 1 swxzxc@126.com
1 College of Life Science and Technology, Mudanjiang Normal University , Mudanjiang, Heilongjiang , China
2 The Second Clinical College, Guangzhou University of Chinese Medicine , Guangzhou, Guangdong , China
3 School of Chinese Materia Medica, Tianjin University of Traditional Chinese Medicine , Tianjin , China
Beddoe Travis
Electronic publication date: 2025 Oct 3
Publication date: 2025
Volume: 13
Electronic Location ID: e19866
Received 2025 Jan 29; Accepted 2025 Jul 17
Copyright: © 2025 Zhong et al.
Copyright year: 2025
Copyright holder: Zhong et al.
License: This is an open access article distributed under the terms of the Creative Commons Attribution License, which permits unrestricted use, distribution, reproduction and adaptation in any medium and for any purpose provided that it is properly attributed. For attribution, the original author(s), title, publication source (PeerJ) and either DOI or URL of the article must be cited.
License URL: https://creativecommons.org/licenses/by/4.0/

Keywords: Artemisia annua, HSP70, Heat stress, Expression pattern, RT-qPCR

Funding: National Natural Science Foundation of China 82204548 Science and Technology Projects in Guangzhou 2023A04J0466 Young Elite Scientists Sponsorship Program from China Association of Chinese Medicine CACM-2022-QNRC2-B30 Forestry Bureau of Guangdong Province 1247246 Traditional Chinese Medicine Bureau of Guangdong Province 20232032 This work was supported by National Natural Science Foundation of China (grant number 82204548), Science and Technology Projects in Guangzhou (grant number 2023A04J0466), the Young Elite Scientists Sponsorship Program from China Association of Chinese Medicine (grant number CACM-2022-QNRC2-B30), Forestry Bureau of Guangdong Province (grant number 1247246), and Traditional Chinese Medicine Bureau of Guangdong Province (grant number 20232032). The funders had no role in study design, data collection and analysis, decision to publish, or preparation of the manuscript.

==============================
Artemisia annua L., a well-known traditional Chinese medicine, is the main source for production of artemisinin, an anti-malaria drug. Heat shock protein 70 (HSP70) plays an important role in plant growth and development as well as in response to biotic and abiotic stresses. While the HSP70 gene family has been characterized in many species, its role in A. annua remains unclear. To investigate the evolutionary relationships, functions, and expression patterns of the A. annua HSP70 (AaHSP70) gene family, we conducted a comprehensive bioinformatics analysis of the HSP70 gene family in the LQ-9 haplotype 0 genome of A. annua. In this study, 47 AaHSP70 genes containing the HSP70 protein structural domain were identified and were unevenly distributed on seven chromosomes, among which, 39 AaHSP70 genes contained 10 identical conserved motifs and eight genes contained varying numbers of seven to nine motifs. Genome collinearity analysis showed that two pairs of genes were duplicated in genome and duplicated segmental duplication (DSD) was the major mode of amplification for this gene family. Cis-acting elements analysis indicated that AaHSP70 was involved in responding to various biotic and abiotic stresses, such as abscisic acid-responsive and defense and stress responsiveness. Gene expression profiling showed that 45 differentially expressed genes (DEGs) of AaHSP70 genes responded differently to heat treatment, of which 12 genes were up-regulated and two genes were up-regulated and then down-regulated. Gene ontology (GO) enrichment showed that two AaHSP70 genes were enriched in the pathway related to reactive oxygen species (ROS). Furthermore, reverse transcription-qPCR (RT-qPCR) experiments confirmed the expression levels of 10 up-regulated genes. This study provides a comprehensive characterization of the HSP70 gene family in A. annua and systematically identifies AaHSP70 genes that were responsive to heat stress, laying the groundwork for further research into the role of the HSP70 gene family in the response of A. annua to abiotic stress.

Introduction

Artemisia annua L., a traditional Chinese medicine plant from the Asteraceae family, is the main source for artemisinin which is widely used in the treatment of malaria (Monroe et al., 2022; Bhattarai et al., 2007). A. annua is highly environmentally adapted and has a wide natural distribution around the world, with more than 70% of strains grown or reported in China (Li et al., 2017). Despite the abundance of artemisinin resources, high diversity of artemisinin contents of A. annua has affected the stable production of artemisinin (Ma et al., 2018). Climate is the main factor affecting the geographical distribution of A. annua (Qin et al., 2018). In China, A. annua mainly grown in southwest areas, such as Sichuan, Yunnan, Guizhou, Chongqing and so on. However, these areas have been seriously affected by extreme climate in recent years, such as high temperature and drought, which seriously affects the stable production of A. annua yield (Liu et al., 2022). Among them, temperature, as an indispensable environmental factor, is very important for the growth and development of A. annua. The study found that the artemisinin content was significantly increased under high temperature treatment at 40 °C, along with a corresponding up-regulation of the expression of related synthase genes and a down-regulation of the expression of synthase genes of the competing pathways, which promoted the biosynthesis of artemisinin (Lu et al., 2018). Therefore, the effect of high temperature on A. annua was significant, which not only increased the content of artemisinin, but also optimized the synthesis pathway of artemisinin by regulating the expression of related genes.

As global climate change intensifies, the frequency and intensity of extreme heat events are increasing, posing a serious threat to agricultural production and natural ecosystems. Plant growth and development, which are foundational to these ecosystems, are profoundly impacted by extreme heat. This impact includes a reduction in photosynthetic efficiency, an increase in transpiration rates, and the disruption of protein structure and function (Hatfield & Prueger, 2015). In response to these stresses, plants have evolved a complex heat stress response (HSR) mechanism. This response is intricately linked to both the generation and accumulation of reactive oxygen species (ROS) and the regulation and activity of heat shock proteins (HSPs) (Kan et al., 2023; Baxter, Mittler & Suzuki, 2014).

Heat shock protein 70 (HSP70), a highly conserved molecular chaperone, plays a critical role in maintaining protein homeostasis and responding to various stresses, including high temperatures (Al-Whaibi, 2011). HSP70 is central to the intracellular molecular chaperone network, involved in protein folding, assembly, transmembrane transport, and regulation of protein activity (Verena et al., 2024). In addition, HSP70 gene family plays an important role in plant response to abiotic stresses, especially in response to high temperature stress. In Capsicum annuum, 21 CaHSP70 genes play a pivotal role in the growth, development, and heat stress response of chili peppers. Ectopic expression of a cytosolic gene, CaHsp70-2, regulated expression of stress-related genes and conferred increased thermotolerance in transgenic Arabidopsis (Guo et al., 2016). In Solanum lycopersicum, HSP70 protein was found to be involved in BAG9-mediated thermotolerance, functioning by protecting photosystem stability and increasing the efficiency of the antioxidant system (Xu et al., 2024). In Solanum melongema, the expression levels of Hsf and Hsp genes (including sHsp, Hsp60, Hsp70 and Hsp100) in heat-tolerant lines of eggplant were significantly higher after heat stress than in heat-sensitive lines under heat stress conditions, which may be the main reason for the heat tolerance of heat-tolerant eggplant (Gong et al., 2021). Overexpression of the HSP70 gene (BcHSP70) in tobacco overexpressed in kale-type oilseed rape seedlings enhances tolerance to heat stress in tobacco (Wang et al., 2016). HSP70 also improves the antioxidant protection of maize leaves induced by drought and high temperature combined stresses (Hu et al., 2010). All of these studies suggest that HSP70 plays an important role in plant response to high temperature stress.

Given the dual role of elevated temperatures in enhancing artemisinin biosynthesis while simultaneously threatening plant survival (Lu et al., 2018), identifying heat-tolerant A. annua varieties is critical for stabilizing artemisinin supply under climate change. However, the molecular mechanisms underlying thermotolerance in A. annua remain poorly characterized. Notably, up to now, the HSP70 gene family has been identified across a variety of plant species, including Arabidopsis (Lin et al., 2001), rice (Jung et al., 2013), sunflower (Ceylan, Altunoglu & Horuz, 2023), maize (Jiang et al., 2021), cotton (Rehman et al., 2020), and pepper (Guo et al., 2016). However, to the best of our knowledge, HSP70 family members have not been characterized in the A. annua genome. In this study, a total of 47 AaHSP70 genes were identified. The expression profiles of the genes under high-temperature stress conditions were analyzed, and the results were verified by reverse transcription-qPCR (RT-qPCR).

Materials and Methods

Plant materials and handling

Plant materials were prepared according to established protocols (Pan et al., 2025). Young leaves of LQ-9 strain were grown in germination medium (Murashige and Skoog medium (MS) 4.43 g/l + Sucrose 30 g/l + Agar 7 g/l + 6-benzylaminopurine (6-BA) 0.5 mg/l + Naphthaleneacetic acid (NAA) 0.06 mg/l) for 7 day, and then transplanted to rooting medium (MS 2.215 g/l + Sucrose 30 g/l + Agar 7 g/l+ Indole-3-butyric acid (IBA) 0.5 mg/l + 0.1 mg/l NAA). The pH range of both media was 6.0–6.3. Plants were incubated in a constant temperature and humidity incubator (LRH-600A-HSE) at 25 °C, 3000lx, and humidity 70% for three days and then heat treated. The 25 °C control group (A1 at 0, 3, 6, 12, 24 h) was sampled without biological replicates due to experimental constraints. However, for the 40 °C heat treatments, two independent biological replicates (A1 transferred to 40 °C and A2 maintained at 40 °C) were utilized at each time point (0, 3, 6, 12, 24 h). At each time point, 3–5 leaves (including young, middle, and old leaves mixed) were selected to ensure sample representativeness. In order to avoid the effects of rhythmic genes to a great extent, samples from treatments were taken at the same time. After the samples were collected, they were wrapped in tin foil immediately and frozen in liquid nitrogen and stored at −80 °C in Eppendorf tubes (Eppendorf, Hamburg, Germany).

Identification of the AaHSP70 gene family

The genome of A. annua LQ-9 haplotype 0, annotation files, and transcriptome data of different tissues of LQ-9 downloaded from the Global Pharmacopoeia Genome Database (GPGD, http://www.gpgenome.com/) were used in this study (Liao et al., 2022). Data were collected as previously described in Pan et al. (2025). To identify the HSP70 genes in the LQ-9 haplotype 0 genome, the annotated HSP70 proteins were scanned against the PFAM database (Pfam 32.0) using PfamScan with an e-value threshold of ≤1e−5 (http://www.ebi.ac.uk/Tools/pfa/pfamscan). Genes containing the HSP70-specific structural domain (PF00012) were considered as candidate genes. To ensure comprehensiveness and avoid missing gene models, publicly available HSP70 protein sequences from the NCBI NR database (https://www.ncbi.nlm.nih.gov/protein) were also used. These sequences, which contained the same HSP70 domain (PF00012), were employed in a BLASTP search against the LQ-9 haplotype 0. Then, all the candidate HSP70 genes were viewed and manually corrected using the Apollo browser (version 2.3.1) (Dunn et al., 2019), and genes that met the following criteria were retained: (1) genes with expression in any tissue as well as gene structural domain coverage greater than 80%, and (2) HSP70 (PF00012) structural domain is present. The conserved domain of HSP70 proteins was detected by the InterPro (https://www.ebi.ac.uk/interpro/). The amino acid number, molecular weight, and isoelectric point of the AaHSP70 protein were calculated using the online website Expasy ProtParam tool (https://web.expasy.org/protparam/). All genes were analyzed for subcellular localization using the online tool Cell-PLoc 2.0 (http://www.csbio.sjtu.edu.cn/bioinf/Cell-PLoc-2/). Phylogenetic analysis of 47 AaHSP70 genes of A. annua was carried out using MEGA 11 (Tamura, Stecher & Kumar, 2021) software, and the phylogenetic tree was constructed by the Neighbor-joining method, with Bootstrap replications set to 1,000. The codon sequences were analyzed using the MEME online program (https://meme-suite.org/meme/) was used to analyze the conserved motifs in the coding proteins, and the number of replications was arbitrary, and the maximum number of motifs was 10. TBtools software (v2. 119) (Chen et al., 2020) ‘Gene Structure View’ function was used to visualize the gene structure. Finally, TBtools software was used to integrate and visualize images of AaHSP70 phylogenetic trees, gene structures, and conserved motifs. ‘Gene Location Visualize’ function in TBtools was used to extract the position information of the target genes of the HSP70 gene family on the chromosome and the length information of the chromosome of A. annua, and the chromosomal localization map of the AaHSP70 gene family was drawn using MG2C (http://mg2c.iask.in/mg2c_v2.1/).

Phylogenetic relationships, collinearity analysis, and promoter cis-acting elements

We obtained all Arabidopsis protein sequences from the TAIR database (https://www.arabidopsis.org/). The phylogenetic tree was constructed by the Neighbor-joining method (NJ) using MEGA11 software (Tamura, Stecher & Kumar, 2021) with 1,000 bootstrap replicates. Subsequently, we further visualized these data on the ITOL (https://itol.embl.de/) platform as per the methods outlined in Pan et al. (2025). The One Step MCScanX integrated into the TBtools software was employed to identify the synteny relationship and duplication pattern of AaHSP70 genes. Gene collinearity analysis was generated using TBtools software and visualized using the ‘Advanced Circos’ function. The criteria for considering two genes as duplicates were: (1) the similarity between two aligned sequences was at least 70% with an e-value <1e−10, and (2) the length of the match covers at least 70% of the average length of the two aligned sequences. Some of the genes identified by ‘MCScanX’ function as collinear were regarded as WGD genes, while other duplicates were classified based on their genomic proximity: those within the 100 kb region were TD, and those over 100 kb or located on different chromosomes were DSD (Lin et al., 2011). The Ka/Ks values for AaHSP70 gene pairs were calculated using the ‘Simple Ka/Ks Calculator’ tool in TBtools. The promoter sequences, located 2,000 bp upstream of the AaHSP70 genes, were analyzed using PlantCARE (http://bioinformatics.psb.ugent.be/webtools/plantcare/html/), and the results were visualized using the ‘Simple Biosequence Viewer’ function in TBtools.

Expression analysis based on RNA-seq data

Total RNA was extracted from A. annua samples with FastPure Plant Total RNA Isolation Kit (Vazyme). Qualified RNA was sequenced using 2 × 150 bp paired-end protocol on the Illumina NovaSeq 6000 platform. Quality control of the corresponding transcriptome data under different treatments was performed using FastaQC (Ward, To & Pederson, 2020). High quality reads were localized to the LQ-9 haplotype 0 genome sequence using HISAT2 (Kim et al., 2019). Expression levels of each gene were calculated using StringTie (Mihaela et al., 2016). Differentially expressed genes (DEGs) among the different samples were identified using DESeq2 (Love, Huber & Anders, 2014), with |log2(fold change)| ≥ 1 and p-value ≤ 1e−6. Hierarchical cluster analysis and TPM (transcripts per million) values of expression levels were performed using the pheatmap package in R (https://cran.r-project.org/package=pheatmap). Gene ontology annotation was performed using eggNOG-mapper (v. 2.1.5) (Cantalapiedra et al., 2021). Gene ontology (GO) enrichment analysis was cconducted using clusterProfiler (Wu et al., 2021), with enrichment results filtered to retain only those with a p-value ≤ 1e−4. The ggplot2 package, known for creating elegant data visualizations using the grammar of graphics, was then utilized to present the results (https://ggplot2.tidyverse.org/).

RNA extraction and RT-qPCR expression analysis

Total RNA was extracted from fresh leaves with FastPure Plant Total RNA Isolation Kit (Vazyme). First strand cDNA was synthesized by HiScript III 1st Strand cDNA Synthesis Kit (+gDNA wiper) (Vazyme, Nanjing, China) according to the manufacturer’s instructions. AaActin was used as a reference gene (Chen et al., 2023). All primers (Table S1) were synthesized by Guangzhou IGE Biotechnology Co., Ltd. (Guangzhou, China). RT-qPCR (Bustin et al., 2009) reactions were performed using the Applied Biosystems ABI 7500 PCR system (ABI, Waltham, MA, USA). The PCR amplification mixture consisted of 1 μl cDNA, 10 μl ChamQ Universal SYBR qPCR Master Mix (Vazyme Biotechnology Ltd., China), 0.4 μl of 10 μM forward and reverse primers, and 8.2 μl of ddH2O. The initial denaturation step of the RT-qPCR reaction was 95 °C for 30 s. 40 cycles of 95 °C for 10 s, and 60 °C for 30 s for annealing. The standard curve was generated using a five-fold dilution gradient of cDNA. Primer amplification efficiencies (90–110%) and standard curve correlation coefficients (R2 > 0.99) were determined. Gene expression data were normalized using the AaActin reference gene. Amplification was performed using the standard curve to calculate the efficiency (E = 10 − 1/slope-1) and correlation coefficient (R2) values. Relative gene expression was calculated using the 2–ΔΔCt method (Livak & Schmittgen, 2001). All reactions included melt curve analysis (continuous monitoring from 60 °C to 95 °C) and three technical replicates. Analysis and graphing of RT-qPCR data was done using GraphPad Prism software (GraphPad Prism 9.1.0).

Results

Identification and characterization of AaHSP70 gene family

A total of 47 AaHSP70 genes were identified in A. annua LQ-9 haplotype 0 and were named AaHSP70_01 to AaHSP70_47 (Figs. 1A and 1B). The number of AaHSP70 genes exons varied from two (AaHSP70_05) to 14 (AaHSP70_15), and the amino acids number ranged from 519 (AaHSP70_24) to 990 (AaHSP70_38) aa, showing great difference in gene structures of this gene family (Fig. 1C and Table S2). Molecular weight of AaHSP70 proteins ranged from 57.00 (AaHSP70_24) to 111.36 (AaHSP70_38) kDa, and the isoelectric point (PI) ranged from 5.04 (AaHSP70_28) to 9.22 (AaHSP70_12). The average hydrophilicity (GRAVY) of all AaHSP70 members was less than zero, indicating that they were hydrophilic protein. The results of subcellular localization prediction showed that most of the genes were located in mitochondrion, and a few genes were located in the cytoplasm, endoplasmic reticulum, chloroplast, and nucleus. Chromosomal mapping revealed that 42 AaHSP70 genes were unevenly distributed on seven chromosomes (Fig. 2), in which chromosome 4 contained the most AaHSP70 genes, with 11, and chromosome 7 have the least, with only one. By analyzing the composition of conserved motifs, we revealed 10 key conserved motifs that vary in length from 21 to 50 amino acids. Thirty-nine of all AaHSP70 proteins demonstrate the integrity of these 10 motifs, whereas other proteins contain seven to nine such motifs. This finding not only highlights the highly conserved nature of the AaHSP70 gene family, but also maps the complex diversity of its protein structures (Fig. 3).

Figure 1 Phylogenetic tree, gene domain and gene structure of AaHSP70 genes.

(A) Phylogenetic tree of AaHSP70 genes, the values greater than 90% of the bootstrap will be retained in the figure. (B) Domain information identified by PfamScan. (C) Gene structures. Blue rectangles represent the coding region and red rectangles represent the non-coding region, thin black lines connecting two exons represent introns.

Figure 2 The position of AaHSP70 genes on chromosomes of A. annua.

The grey bars signify chromosomes, with its corresponding name labeled Chr1-7. The scale situated on the left side provides a measure of the genetic distance. The AaHSP70 gene family members are marked on chromosomes with blue letters.

Figure 3 Gene structures and protein motifs of AaHSP70 gene family in A. annua.

(A) Conservative motif analysis of AaHSP70 proteins. The colorful boxes delineated different motifs. (B) The sequence logos for each conserved motif.

Analysis of collinearity and duplication types of AaHSP70 genes

Gene duplication can occur by a variety of mechanisms, including whole-genome duplications (WGD) type, tandem duplication (TD) type, and duplicated segmental duplication (DSD) type (Zheng et al., 2024). A total of 63 duplicated gene pairs were identified within 47 AaHSP70 genes. Collinearity analysis of 42 AaHSP70 genes was conducted within species, and the results showed that two pairs of genes showed collinearity, namely AaHSP70_03 and AaHSP70_11, AaHSP70_33 and AaHSP70_41, respectively, indicative of WGD (Fig. 4). Of the total number of duplicated gene pairs found throughout the genome, there was a clear distribution of duplication types: two pairs of genes were categorized as WGD, accounting for 3.2% of the total. In addition, nine pairs of genes were categorized as TD, accounting for 14.3% of the total. The majority of these duplicated gene pairs were categorized as DSD, with 52 pairs (82.5% of the total), indicating that DSD was the major gene duplication type in the AaHSP70 gene family (Table S3).

Figure 4 Collinearity analysis of the AaHSP70 gene family.

This circular diagram illustrates the chromosomal distribution and collinearity relationships of AaHSP70 genes in A. annua. The outer ring represents the different chromosomes (chr1 to chr9) of A. annua. Grey lines connect genes that have undergone whole-genome duplication events, indicating their paralogous relationships. Red lines specifically highlight the collinear AaHSP70 genes, showing syntenic blocks within the genome.

In addition, by analyzing the Ka/Ks values of duplicate gene pairs in the AaHSP70 gene family, we found that 58 gene pairs had Ka/Ks ratios less than one (Table S3), indicating that these AaHSP70 genes were in the purifying selection stage and tended to eliminate deleterious mutations and maintain functional stability.

Phylogenetic relationship of HSP70 genes between A. annua and A. thaliana

In exploring the lineage-specific expansion of the AaHSP70 gene family, the phylogenetic tree of the HSP70 genes in A. annua and A. thaliana was conducted (Fig. 5). The phylogenetic tree was divided into three clades, representing three subfamilies, in which the first subfamily containing 12 HSP70 genes, the second subfamily expanded this number to 20, while the third subfamily was the largest, comprising 51 HSP70 genes. The variation in gene count across the subfamilies suggested a divergence in functional roles and distinct evolutionary paths within the HSP70 gene family. Notably, there were 25 AaHSP70 forming a distinct cluster in subfamily III, signifying their status as a highly conserved group. This conservation likely underscores the genes stable presence throughout evolution, coupled with their pivotal roles in a variety of species.

Figure 5 The figure illustrates the phylogenetic relationships of the HSP70 gene family in A. annua and A. thaliana.

This circular phylogenetic tree depicts the evolutionary relationships among HSP70 genes from A. annua and A. thaliana. Each branch corresponds to a specific gene, and the nodes indicate common ancestors. The numbers on the branches represent Bootstrap support values, with only values above 90% being displayed. The tree is partitioned into three main groups (I, II, III), reflecting the classification of HSP70 genes into different subfamilies based on their evolutionary divergence and functional specialization.

Cis-regulatory elements analysis of AaHSP70 genes promoters

To better understand how AaHSP70 participated in the regulation of biotic and abiotic stresses, the 2,000 bp upstream promoter sequences of the 46 AaHSP70 genes were used to search the cis-regulatory. Elements using PlantCARE online software (Fig. 6). Many cis-acting elements involved in hormone response were detected in the promoter region of the AaHSP70 genes, such as abscisic acid-responsive element (ABRE), auxin-responsive element (TGA-element), gibberellin-responsive element (P-box, TATC-box), Methyl Jasmonate-responsive element (CGTCA-motif, TGACG-motif) and salicylic acid response element (SARE, TCA-element). In addition, many stress response elements, such as the drought response element (MBS), the low-temperature response element (LTR), the defense and stress responsiveness response element (TC-rich repeats), and the G-box, were detected to be associated with the light stress response of plants. The above analysis suggested that the AaHSP70 genes contained a large number of cis-acting elements in response to biotic and abiotic stresses. In addition, we found significant heterogeneity in the distribution of cis-acting elements. For example, ABRE elements were identified in the promoter regions of 44 genes, whereas SARE elements appeared in only two genes, which may reflect the complexity and specificity of gene expression regulation.

Figure 6 Analysis of Cis-regulatory elements in AaHSP70 genes promoters.

This figure displays the distribution of various cis-regulatory elements within the promoter regions (2,000 bp upstream) of AaHSP70 genes. Each row corresponds to a specific AaHSP70 gene, and the different colored dots represent distinct types of cis-regulatory elements, as listed in the legend on the right. The absence of a dot indicates that the corresponding cis-element is not present in the promoter region of that gene. The numbers on the x-axis denote the positions relative to the transcription start site (TSS), with 0 representing the TSS and negative values indicating upstream positions.

Expression of AaHSP70 genes in response to heat stress and GO enrichment analysis

To further investigate the gene expression changes in A. annua under heat treatment, a series of heat treatment experiments were conducted on A. annua and transcriptome sequencing was performed. A total of 3.95 to 5.37 Gb raw data was obtained for these samples (Table S4). In this study, A. annua plants were treated at a high temperature of 40 °C for specific time intervals: 0, 3, 6, 12, and 24 h. Each of these time points at the elevated temperature was compared against a corresponding control group at the optimal growth temperature of 25 °C, also at 0, 3, 6, 12, and 24 h, to evaluate the effects of heat stress on plant physiology and gene expression. Differential expression analysis revealed that a total of 6,551 DEGs were identified compared to control group, of which 2,720 were up-regulated and 3,831 were down-regulated genes. Among them, 45 AaHSP70 genes were included, and the expression profiles of these 45 AaHSP70 genes under different treatments were mapped (Fig. 7A). Compared with the control group, 12 genes (AaHSP70_03, AaHSP70_09, AaHSP70_11, AaHSP70_16, AaHSP70_25, AaHSP70_26, AaHSP70_27, AaHSP70_31, AaHSP70_40, AaHSP70_43, AaHSP70_44, AaHSP70_45) were significantly up-regulated and two genes (AaHSP70_28 and AaHSP70_29) were first up-regulated and then down-regulated.

Figure 7 AaHSP70 expression profiles and up-regulated genes GO enrichment maps.

(A) Expression patterns of 45 AaHSP70 genes. The rows of the heat map represent different members of the AaHSP70 gene family, and the columns represent samples at different time points (0, 3, 6, 12, 24 h) or under different treatment conditions (25 °C and 40 °C). The color scale indicates the range of Log10(TPM+1), with red representing higher expression levels and blue representing lower expression levels. All data were normalized by row to allow for comparison across different genes and conditions. (B) GO enrichment of all up-regulated. Each point in the graph represents a GO entry, the horizontal coordinate is the Gene Ratio, and the vertical coordinate lists the enriched GO entries. The color indicates the adjusted p-value, the redder the color, the smaller the p-value, the more significant the enrichment.

GO enrichment of the 2,720 up-regulated genes showed that 46 DEGs were categorized as ‘cellular response to heat’ and 22 DEGs were categorized as ‘obsolete positive regulation of transcription from RNA polymerase II promoter in response to heat stress’. In addition, 59 DEGs were enriched in ‘response to hydrogen peroxide’ and 41 DEGs were enriched in ‘response to reactive oxygen species’. Interestingly, two of the up-regulated genes (AaHSP70_25 and AaHSP70_27) also were enriched in both pathways (Fig. 7B and Table S5). It was hypothesized that these two genes may have a protective effect on plants by regulating ROS levels under high temperature conditions.

RT-qPCR analysis of the AaHSP70 gene family

To confirm the accuracy of the transcriptomic data, we performed RT-qPCR validation on selected genes among the 12 up-regulated AaHSP70 genes that showed significant differential expression in RNA-seq analysis, including AaHSP70_03, AaHSP70_09, AaHSP70_11, AaHSP70_25, AaHSP70_27, AaHSP70_31, AaHSP70_40, AaHSP70_43, AaHSP70_44, and AaHSP70_45. RT-qPCR was performed at 0, 3, and 24 h after heat treatment (Fig. 8 and Table S6). The RT-qPCR results were in excellent agreement with the RNA-seq data, strongly demonstrating the increased expression of these genes under heat stress. In particular, AaHSP70_25 and AaHSP70_27, which were distinguished in GO enrichment analysis for their role in the reactive oxygen species response, were verified to be significantly upregulated. In conclusion, RT-qPCR analysis confirmed the transcriptome findings, identifying that the 10 AaHSP70 genes under study were indeed upregulated upon heat treatment at 40 °C. This validation not only strengthened the transcriptomic data, but also laid the foundation for subsequent functional analyses.

Figure 8 Relative expression levels of ten AaHSP70 genes under 40 °C heat treatment.

NS’ indicates no significance. Ten gray bars indicate control group, red and blue bars indicate heat-treated group. Asterisks (*, **, ***, ****) represent significant differences of P < 0.05, P < 0.01, P < 0.001 and P < 0.0001, respectively. The figure displays the mean values and standard deviations of three replicates. 0, 3, and 24 h represent the three time points of heat treatment, respectively.

Discussion

Rising global temperatures profoundly affect plant growth, development, and reproduction (Lobell, Schlenker & Costa-Roberts, 2011; Pauli et al., 2012) and are greatly threatening global crop yields. HSP70 is critical for the response to biotic and abiotic stresses. The HSP70 gene family has been identified in several plant species in recent years. The HSP70 gene family has been identified in several plant species in recent years. However, the HSP70 gene family has not been reported in A. annua.

In this study, a total of 47 AaHSP70 genes were identified from LQ-9 haplotype 0 genome of A. annua. The number of AaHSP70 genes were more than maize (22 members) (Jiang et al., 2021), Capsicum chinense (20 members) (Ding et al., 2021), and Litchi chinensis (18 members) (Fan et al., 2024), but less than in sunflower (103 members) (Pang et al., 2019), and soybeans (61 members) (Zhang et al., 2015b). The majority of AaHSP70 genes were found to have amino acid sequences in the range of 500 to 600 amino acids. Comparative analysis revealed a similarity in the number of introns and the structure of the phylogenetic tree among these genes. This suggests that there is a correlation between the gene structures of closely related AaHSP70 genes. Analysis of conserved motifs showed that 39 genes contained 10 conserved motifs, and eight genes contained seven to nine different numbers of motifs. This indicates that the AaHSP70 gene family was very conserved. This is similar to the HSP70 gene family found in Chrysanthemum (Mengru et al., 2023). Five genes were not assembled to the chromosomes, and the 42 genes were unevenly distributed across the seven chromosomes. The diversity of plant genes relies on events of gene duplication, a process that is a key evolutionary mechanism for the expansion of gene families and provides the possibility for the divergence of gene functions (Das et al., 2016; Das Laha et al., 2020). The dominance of DSD in AaHSP70 gene family expansion (82.5%) suggests adaptive evolution under environmental pressures. This may facilitate functional diversification of HSP70 chaperones in artemisinin biosynthesis pathways. Collinearity analysis of 42 genes within the species revealed two gene pairs, which may indicate that they originated from duplications of the same ancestral gene. Additionally, 58 duplicated gene pairs exhibited Ka/Ks values less than one, suggesting that these genes have been subjected to purifying selection (Hurst, 2002), which may be related to the relatively stable function of this gene family in responding to abiotic stress.

Phylogenetic analysis showed that the A. annua HSP70 gene family formed a distinct clustering with the A. thaliana HSP70 gene family, indicating that these genes were highly conserved during evolution. This is consistent with the findings of A. thaliana, which revealed that members of the HSP70 gene family were widely conserved in plants (Lin et al., 2001). This conservation may reflect the stability and functional importance of these genes across species, and may also indicate their common role in the evolution of plant adaptations. Cis-acting elements play important roles in the regulation of many processes, including plant growth, development, and stress response. Analysis of the AaHSP70 genes promoter revealed the presence of many cis-acting elements in the promoter region associated with abiotic stress responses, including ABRE, CGTCA-motif, TC-rich, TCA-element, and LRT, suggesting that the AaHSP70 genes may be associated with different abiotic stresses, with the highest proportion of ABRE. Studies have shown that heat tolerance of plants can be improved by exogenous use of abscisic acid (ABA) (Huang et al., 2016). Under high-temperature stress, ABA treatment of tall fescue up-regulated the expression of FaHSFA2c and ABA-responsive transcription factors (FaAREB3 and FaDREB2A), suggesting that these factors may regulate the transcriptional activity of FaHSFA2c, thereby enhancing heat tolerance (Wang et al., 2017).

HSP70, as a key member of the heat shock protein family, has garnered significant attention due to its high abundance and strong inducibility within cells. It not only protects cells from stress-induced damage and promotes the repair of damaged cells but also possesses important biological functions such as anti-inflammatory, anti-apoptotic, and tolerance to ischemia-hypoxia damage (Bao et al., 2015). Additionally, A. annua is rich in secondary metabolites with anti-inflammatory activity, such as artemisinin, flavonoids, and phenolic acids. The anti-inflammatory effects of A. annua may result from the synergistic action of these secondary metabolites with HSP70 proteins (Abate et al., 2021). Secondary metabolites may alleviate oxidative damage and inhibit inflammatory signaling pathways, synergizing with the molecular chaperone function of HSP70 to jointly regulate inflammatory responses, thereby enhancing the overall stress resistance of A. annua. Studies have shown that Artemisia sieberi, a closely related species of A. annua, prefers to grow on south-facing slopes, while A. annua exhibits increased artemisinin production under high-temperature conditions (Lu et al., 2018; Mirdavoudi et al., 2022). These findings suggest that strong light and high temperatures may be key inducing factors for secondary metabolism in Artemisia species. Specifically, the strong light and high-temperature environment of south-facing slopes may provide important environmental triggers for the synthesis of secondary metabolites in Artemisia species, thereby influencing their secondary metabolic processes and product accumulation. In this study, the gene expression pattern of A. annua leaves at different time points (0, 3, 6, 12 and 24 h) under heat treatment at 40 °C was thoroughly analyzed. A total of 6,551 DEGs were identified by means of RNA-seq analysis, including 45 AaHSP70 genes. Notably, two genes, AaHSP70_17 and AaHSP70_18, were not detected to be significantly expressed in leaf tissues, suggesting that they may not be involved in daily leaf function or stress response. In contrast, AaHSP70_17 and AaHSP70_18 showed high expression levels in flowers, which may indicate that they play key roles in flower development, during reproduction, or in response to specific environmental signals. There were 12 up-regulated genes, six of which were significantly higher in expression at the 3 h treatment, possibly reflecting a reduction in plant dependence on these genes under sustained heat stress or plant adaptation to heat stress through other regulatory mechanisms. AaHSP70_03 and AaHSP70_09, show a continuous increase in expression levels with prolonged heat treatment. This sustained up-regulation suggests they may play a crucial and ongoing role in the heat stress response, potentially contributing to the stabilization of cellular proteins and membranes under high temperatures. Conversely, genes like AaHSP70_28 and AaHSP70_29 exhibit an initial up-regulation followed by down-regulation. This pattern may indicate a transient response to heat stress, where these genes are pivotal in the immediate defense against heat-induced damage but their expression diminishes as the plant acclimates to the stress conditions. This finding echoes several recent studies on the response of plants to heat stress. For example, the study found a similar phenomenon in poplar, where the HSP70 gene was rapidly up-regulated after heat stress, highlighting the critical role of these genes in plant heat stress response (Zhang et al., 2015a). Similarly, the study observed in Chrysanthemum an increase in the expression of HSP70 genes in response to heat stress, correlating with heat tolerance in plants (Yin et al., 2023). In addition, the study in tomato also showed a link between HSP70 gene expression and heat tolerance (Xu et al., 2024). Together, these studies emphasize the importance of the HSP70 gene family in plant thermoprotective mechanisms, and provide a valuable background for further studies on the function of these genes in A. annua. The GO enrichment of all up-regulated DEGs revealed that 68 up-regulated DEGs were enriched in pathways associated with heat stress. Also, two AaHSP70 genes (AaHSP70_25 and AaHSP70_27) were enriched in pathways related to reactive oxygen species (ROS). They may protect antioxidant enzymes (such as catalase) through molecular chaperone-mediated protective effects, thereby alleviating heat-induced oxidative stress. This protective effect may help maintain intracellular redox balance and reduce oxidative damage. Additionally, their upregulation may enhance cellular adaptability to heat stress by promoting the repair of damaged proteins and preventing protein aggregation, further mitigating the negative effects of heat stress on cellular function. Under heat stress conditions, the accumulation of ROS typically leads to oxidative stress, which damages cell membranes, proteins, and DNA. The molecular chaperone functions of AaHSP70_25 and AaHSP70_27 may be crucial for maintaining the activity and stability of antioxidant enzymes. By protecting these enzymes, they may help enhance the plant’s antioxidant defense system, thereby improving the plant’s survival capacity under high-temperature conditions. Additionally, the expression patterns of these two genes show that they are rapidly upregulated shortly after heat treatment and then gradually decline, suggesting that they may play a key role in the early response to heat stress, helping plants quickly adapt to environmental changes (Ahmad, Sarwat & Sharma, 2008; Driedonks et al., 2015).

Therefore, in this study, 47 AaHSP70 genes and their gene structures, conserved motifs, chromosomal distribution and cis-acting elements were identified by bioinformatics analysis. The study also identified genes with significant changes in expression levels under heat stress conditions through RNA-seq data analysis. The analytical results of this study provide a good foundation for further in-depth study of HSP70 function.

Conclusion

In this study, we present the first comprehensive and systematic analysis of the HSP70 gene family in the A. annua. We identified 47 AaHSP70 genes from the genome of A. annua LQ-9 haplotype 0 and analyzed their phylogenetic relationships, conserved structural domains, gene structures, and gene duplication events. Gene duplication events indicated that DSD is the major mode of amplification for the AaHSP70 gene family. In addition, the phylogenetic tree between the HSP70 genes of A. annua and A. thaliana showed that AaHSP70 gene family was significantly conserved. The cis-regulatory elements indicated that the promoters of AaHSP70 genes contain a large number of biotic and abiotic stress elements. The expression patterns and GO enrichment of AaHSP70 genes at 0, 3, 6,12 and 24 h of heat stress treatment indicated that these genes were regulated under heat stress and predicted the potential functions of these genes through multiple pathways. The results of the study revealed the characteristics of the AaHSP70 gene family, laid the foundation for further research on the AaHSP70 gene family and its ability to regulate plant responses to environmental stress, and also provided theoretical guidance for breeding heat-tolerant aniseed varieties and agricultural production.

Supplemental Information

Supplemental Information 1 Primers used in gene expression validation.

Supplemental Information 2 The basic information about the HSP70 genes in A. annua LQ-9 haplotype 0.

Supplemental Information 3 Duplication types, Ka/Ks ratio calculations.

Supplemental Information 4 Statistic of RNA-seq data.

Supplemental Information 5 DEGs enriched to heat stress and reactive oxygen pathways in GO enrichment.

Supplemental Information 6 Relative experssion levels of ten AaHSP70 genes and experimental methodology.

Supplemental Information 7 MIQE Checklist.

Additional Information and Declarations

Competing Interests

The authors declare that they have no competing interests.

Author Contributions

Shan Zhong conceived and designed the experiments, performed the experiments, analyzed the data, prepared figures and/or tables, authored or reviewed drafts of the article, and approved the final draft.

Hengyu Pan conceived and designed the experiments, analyzed the data, prepared figures and/or tables, authored or reviewed drafts of the article, and approved the final draft.

Chaoxue Ma performed the experiments, analyzed the data, prepared figures and/or tables, and approved the final draft.

Haojia Xu analyzed the data, prepared figures and/or tables, and approved the final draft.

Xiaoxia Ding performed the experiments, authored or reviewed drafts of the article, and approved the final draft.

Shengye Bao analyzed the data, prepared figures and/or tables, and approved the final draft.

Siyu Zhao performed the experiments, prepared figures and/or tables, and approved the final draft.

Peiqi Shi performed the experiments, prepared figures and/or tables, and approved the final draft.

Baosheng Liao conceived and designed the experiments, analyzed the data, authored or reviewed drafts of the article, and approved the final draft.

Xianchun Zong conceived and designed the experiments, analyzed the data, authored or reviewed drafts of the article, and approved the final draft.

Data Availability

The following information was supplied regarding data availability:

Transcriptome data for LQ-9 plants of Artemisia annua under heat treatment are available at the Global Pharmacopoeia Genome Database: http://www.gpgenome.com/species/92.

Raw data are available in the Supplemental Files.

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
