# Peer review of "Genome-wide identification and expression analysis of the HSP70 gene family in Artemisia annua L. under heat stress"

_PeerJ, doi:10.7717/peerj.19866_

## Round 0.1 · original submission · Major Revisions

Please address all concerns raised by the reviewers.

Reviewer 1 ·

Basic reporting

- Line 21: Please check the full name in Latin - Artemisia annua L. (check the entire manuscript)
- Asteraceae - italic writing
- I suggest in the introduction that the reason for choosing the Annua species be more clearly stated.
- Line 350: A. annua leaves or stems?

Experimental design

-

Validity of the findings

-

Reviewer 2 ·

Basic reporting

The manuscript entitled ‘Genome-wide identification and expression analysis of the HSP70 gene family in Artemisia annua L. under heat stress’ presents a comprehensive genomic analysis of the HSP70 gene family in Artemisia annua, a plant known for the production of artemisinin, an important antimalarial drug. The main aim of the study is to identify, characterise and analyse the expression profile of these genes in response to heat stress.
The manuscript is generally well written, with clear scientific language and vocabulary appropriate to the discipline. However, the English needs stylistic revision in several places to improve fluency and readability, especially in some sentences in the introduction and the ‘Discussion’ section, where the repetition of concepts already expressed (e.g. function of HSP70, number of genes, evolutionary conservation) is redundant. I suggest professional linguistic revision.
The structure is coherent and complete, but the introduction is too long and repetitive: many paragraphs dwell on general concepts related to HSP70 that are already known and mentioned several times. Greater conciseness would be desirable, with a clearer focus on the specific relevance of the study in A. annua, especially considering its role in artemisinin biosynthesis. The authors should provide more information on biological replicates and statistical analysis. Greater methodological transparency is needed.

Experimental design

1. The ‘Materials and Methods’ section is detailed, but lacks clarity on how many biological replicates were used for each treatment. This is crucial to ensure the statistical robustness of the data.
2. It is not specified whether the heat treatment was replicated in several independent experiments, nor the number of plants per sampling point. This reduces confidence in the generalisability of the results.
3. The experimental design involves two groups (A1 and A2), but the distinction between the two is not clearly explained in the results. The management of controls at 25°C and conditions at 40°C for different times should be better clarified.

Validity of the findings

4. The biological interpretation of the data remains somewhat superficial, with generic statements and no real functional insight into the up-regulated genes.
5. The GO enrichment analyses are described correctly, but the results are only mentioned briefly in the text. A more in-depth discussion of the specific role of the AaHSP70_25 and AaHSP70_27 genes, for example, would be appropriate, considering their relevance to the response to ROS.
6. Validation by RT-qPCR is commendable, but is reported too succinctly. It should be specified whether efficiency values were calculated for each primer and whether the data were normalised correctly.

Additional comments

7. The discussion is too descriptive and lacks real insight into the biological implications of the results. In this regard, in order to enrich the discussion, I suggest that authors discuss several recent manuscripts: Phytochemical analysis and anti-inflammatory activity of different ethanolic phyto-extracts of artemisia annua L., DOI: 10.3390/biom11070975; Ecological Niche Modelling and Potential Distribution of Artemisia sieberi in the Iranian Steppe Vegetation, DOI: 10.3390/land11122315

8. Overlap between introduction and discussion: too many repetitions of concepts already expressed (e.g. evolutionary conservation of HSP70, effect of heat on A. annua).
9. Some figures (e.g. heat map, phylogenetic tree) are complex and need to be made more readable and accompanied by more explanatory legends.

---

## Round 0.2 · accepted · Accept

The authors have addressed all the reviewers' comments.

Reviewer 2 ·

Basic reporting

The authors have responded to all the critical issues raised by all the reviewers. In my opinion, the manuscript can be accepted.

Experimental design

The authors have responded to all the critical issues raised by all the reviewers.

Validity of the findings

The authors have responded to all the critical issues raised by all the reviewers.

Additional comments

NA